

# The crustal structure of the Lomgmenshan fault zone and its implications for seismogenesis: New insight from aeromagnetic and gravity data

Hai Yang[1,2,3], Shengqing Xiong[1,3], Qiankun Liu[1], Fang Li[1], Zhiye Jia[1], Xue Yang[1],
Haofei Yan[1], Zhaoliang Li[1]

[1]China Aero Geophysical Survey and Remote Sensing Center for Natural Resources, Beijing, 100083
[2]State Key Laboratory of Lithospheric Evolution, Institute of Geology and Geophysics, Chinese Academy of Sciences, Beijing, 100029
[3]Key Laboratory of Airborne Geophysics and Remote Sensing Geology, Ministry of Natural Resources,
Beijing 100083, China

*Correspondence to*: Shengqing Xiong (xsqagrs@126.com); Hai Yang (yanghai@mail.cgs.gov.cn)

**Abstract.** Although many geophysical models have been proposed in the Longmenshan fault zone (LFZ) and its surrounding areas, the deep structure of the seismic gap and its constraint of the Wenchuan and Lushan earthquakes remain uncertain. Based on the compiled aeromagnetic data and Bouguer gravity data, we have tried to create a more detailed and visible magnetic and density model beneath the LFZ using 2D forward modeling and 3D inversion. The research shows that structure heterogeneities are widely distributed beneath the LFZ. The earthquake epicenters show high magnetic anomalies and the edge of high Bouguer gravity anomalies that consist of rigid blocks where apt to accumulate stress. However, the seismic gap shows low magnetic anomalies and transition of Bouguer gravity anomalies related to a weak zone. The Sichuan Basin has two NE-trending banded high magnetic blocks extending beneath the LFZ that firmly support the crust of the Sichuan Basin was downward subduction toward the LFZ. More importantly, the basement subducts to approximately 33 km west of the Wenchuan-Maoxian fault with a low dip angle beneath the middle segment of the LFZ, whereas the distance decreases to approximately 17 and 19 km under the southern segment. Thus, the crust of the Sichuan Basin beneath the middle segment extends farther than that beneath the southern segment with the seismic gap as the transition zone. Therefore, we propose that the structural heterogeneity of the basement on the western margin of the Sichuan Basin may be the main reason for the different focal mechanisms and geodynamics of the Wenchuan and Lushan earthquakes.



## 1 Introduction

In 2008 and 2013, the devastating Wenchuan ($M_s$ 8.0) and Lushan ($M_s$ 7.0) earthquakes successively struck the LFZ in the eastern margin of the Tibetan Plateau. The two earthquakes caused great losses of human lives and properties in China. Many researchers focus on the mechanism of two earthquakes because they occurred at a close distance with distinct fault geometry, surface rupture, coulomb stress, and deep structure (Chen et al., 2013; Li et al., 2013, 2014; Lei and Zhao, 2010; Pei et

al., 2010; Shan et al., 2013; Wang et al., 2009, 2014a, 2015, 2017a; Wu et al., 2013; Zhan et al., 2013; Zhao et al., 2013). For instance, the Wenchuan earthquake occurred on the Yingxiu-Beichuan fault in the middle segment of the LFZ, while the Lushan earthquake occurred on a blind reverse fault in the southern segment (Chen et al., 2013; Li et al., 2014; Wang et al., 2014a). The Wenchuan earthquake occurred on a thrust fault associated with dextral strike-slip movement and had a surface rupture extending more than

300 km toward the NE. However, the Lushan earthquake was dominated by concealed thrust fault with the rupture zone restricted to 30 km underground and no obvious surface rupture (Chen et al., 2013; Zhao et al., 2013). It is noted that there is a 40~60 km gap void of aftershocks for both earthquakes along the fault zone between the Wenchuan and Lushan earthquakes. (Du et al., 2013; Liu et al., 2018; Pei et al., 2014; Wang et al., 2015; Zhu et al., 2016).

The LFZ was attacked by two different earthquakes and the risk of the seismic gap has challenged Earth scientists. In the past decade, numerous studies have been carried out on the high-resolution deep structure of the LFZ. It is generally accepted that the deep structure of the 2013 Lushan hypocenter is characterized by a high velocity (Vp, Vs), low Poisson's ratio, and high resistivity, whereas high velocity (Vp, Vs), high Poisson's ratio, and high resistivity were imaged at the 2008 Wenchuan hypocenter (Li et

al., 2013; Lei and Zhao, 2010; Pei et al., 2010; Wang et al., 2009, 2015; Zhan et al., 2013). The gap area between two earthquakes is characterized by low velocities (Vp, Vs), high Poisson's ratios, and high conductivity (Liu et al., 2018; Pei et al., 2014; Wang et al., 2014a, 2015, 2017b; Zhan et al., 2013). Several models recently proposed by different research groups to explain the formation of a low-velocity seismic gap, (1) a weak and ductile deformation area formed by the strong compression between the

Tibetan Plateau and the Sichuan Basin (Pei et al., 2014), (2) a fault zone caused mantle upwelling and partial melting (He et al., 2017; Liang et al., 2018), (3) fluid-bearing ductile flow from the mid-lower



crust of Tibet (Wang et al., 2014a). Therefore, the formation of the low velocity and high-conductivity gap area beneath the seismic gap still cannot reach an agreement, because the large-scale crustal shortening resulted in complex deep structure under the LFZ.

Previous interpretation of the magnetic and gravity data trying to describe the magnetic and density differences along the LFZ. 1) the Wenchuan and Lushan earthquakes were separated by NNW-trending (near EW) high magnetic material, the Wenchuan earthquake distributed in the northeast segment of the negative magnetic anomaly belt (Yan et al., 2016), 2) the LFZ is characterized by large negative magnetic anomaly area caused by some crystalline complexes with reversal magnetization in the crust (Yan et al.,

2016; Zhang et al., 2010), 3) the LFZ lies in the gravity gradient zone that is related to the density discontinuity in deep (Chen et al., 2013; Zhang et al., 2010). However, the lack of detailed magnetic and gravity structure makes it difficult to understand the deep structure of the LFZ. To create an integrated geophysical model beneath the middle and southern segments of the LFZ, in this study we conducted a quantitative analysis using 2D forward modeling under the constraint of velocity model and 3D inversion.

Then, we discuss the structure geometry and physical property under the LFZ and adjacent two blocks, which is essential to advance our understanding on the genesis of the seismic gap and geodynamics of two earthquakes.

## 2. Tectonic setting

The LFZ lies in a convergent area between the Songpan–Ganzi block (SGB) and the Sichuan Basin

at the eastern margin of the Tibet Plateau (Fig. 1a). It is characterized by a steep topographic gradient, complex strike-slip, and strong thrust motions with an average elevation rising 4000m above the Sichuan Basin over a distance of less than 100 km (Burchfiel et al., 2008; Chen and Wilson, 2008; Guo et al., 2013; Parsons et al., 2008; Royden et al., 2008). The fault zone is bounded by the Wenchuan - Maowen fault in the west, and the Guanxian-Anxian fault in the east. The Mian-Lue suture formed the northeast

boundary of the fault zone and connected with the Kangdian tectonic belt to the southwest. The SGB is covered by a 10 km Middle to Upper Triassic flysch formation in the northwest of the LFZ. Triassic syn-tectonic adakitic-type granitoids are widely distributed in the SGB, which are likely sourced from the partial melting of an underlying Proterozoic basement that is part of the Sichuan Basin (Dai et al., 2011;



Hu et al., 2005; Roger et al.,2010; Zhang et al.,2006; Zhao et al., 2007a, b) (Fig.1b). The Sichuan Basin

has a Precambrian crystalline basement overlain by late Proterozoic to Cenozoic sedimentary cover.

Geological and geophysical studies suggest the crust of the Sichuan basin probably extends beneath the

LFZ (Burchfiel et al., 1995; Guo et al., 2013; Roger et al., 2010; Wang et al., 2015; Zhang et al., 2006;

Zhu et al., 2008).

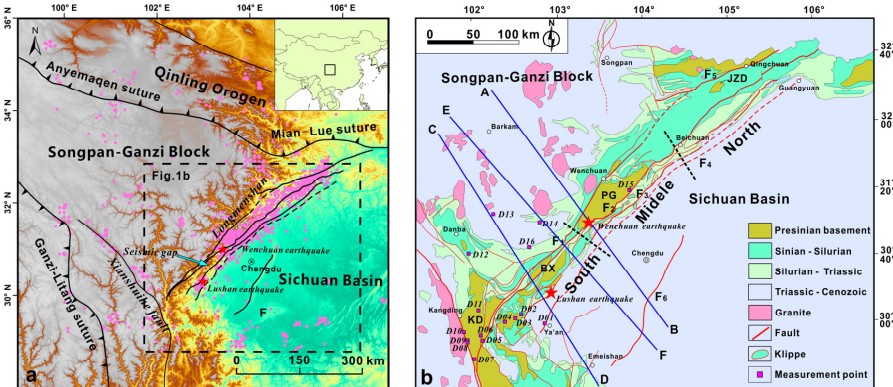

**Figure 1.** Geological background of the LFZ. (a) Tectonic map of the LFZ and adjacent area. (b) Geological map of

the LFZ, showing the location of modeling profiles as blue lines (modified after Li et al. (2008) and Yan (2011)).

$F_1$: Wenchuan-Maoxian fault, $F_2$: Yinxiu-Beichuan fault, $F_3$: Guanxian-Anxian fault, $F_4$: Guangyuan-Dayi concealed

fault, $F_5$: Pingwu-Qingchuan fault, $F_6$: Longquanshan fault; PG: Pengguan complex; BX: Baoxing complex; KD:

Kangding complex; JZD: Jiaoziding complex. Red stars show the 2013 Lushan and 2008 Wenchuan earthquakes,

respectively. Pink dots indicate the 1204 earthquakes (earthquake magnitude≥4) from 1970 to 2014.

There are four reversed thrust and strike-slip faults from northwest to southeast, including the

Wenchuan-Maowen fault ($F_1$: rear fault), the Yingxiu-Beichuan fault ($F_2$: central fault), the Guanxian-

Anxian fault ($F_3$: fore fault), and the Guangyuan-Dayi fault ($F_4$: range front blind fault) (Fig. 1b). The

Wenchuan earthquake occurred in the Yingxiu-Beichuan fault. The Lushan earthquake occurred in a blind

reverse fault between the Guanxian-Anxian fault and the Guangyuan-Dayi fault in the southern segment

of the LFZ (Chen et al., 2013; Li et al., 2013; Wang et al., 2014b). Due to the distinct deformation style,

evolution history, and foreland sediment, the LFZ can be divided into three segments along the strike

(Fig. 1b). The northern segment has an imbricated thrust fault system in front of the Jiaoziding (JZD)

complex and the Tangwangzhai syncline. The middle segment outcrops the Pengguan (PG) complex with

Proterozoic to Triassic klippes developing in front of it. The southern segment outcrops the Baoxing (BX)



complex with Silurian to Triassic klippes distributing in front of it (Li et al., 2008). Unlike the PG complex is fault contact with later formations, the BX complex is covered by a successive formation showing parallel unconformity or conformity. From the northeast to southwest of the LFZ, the deformation age is getting younger, the deformation is more brittle and extensive, and the Cenozoic

activity is more intense (Li et al., 2008). The Wenchuan earthquake was located in the middle segment of the LFZ, while the Lushan earthquake occurred in the southern segment. The seismic gap lies in the transition zone between the middle and southern segments.

### 3 Data and Method

### 3.1 Rock magnetic susceptibility measurement

The Proterozoic to Cenozoic strata are successively distributed in the LFZ and surrounding area. Measuring the magnetic susceptibility of these formations is important for the interpretation of aeromagnetic anomalies. Therefore, a field survey of rock magnetic susceptibility was conducted in the LFZ, especially the high aeromagnetic anomaly area. Each type of lithology test 30 points for reliable statistic results using a KM-7 Magnetic Susceptibility kappameter (Fig. 1 and Table 1). The results show

that Proterozoic serpentine has the highest magnetism with an average magnetic susceptibility value of 0.0765 SI. Proterozoic quartz diorite has a moderate magnetic susceptibility of 0.0238 - 0.0487 SI with an average value of 0.0377 SI, while Proterozoic granite has values of 0.0002 - 0.0247 SI with an average value of 0.0068 SI. The Triassic and Jurassic granites are widely distributed in the west of the LFZ, and their magnetic susceptibility ranges from 0.0118 - 0.0201 SI, and their average value is 0.0167 SI. The

magnetic susceptibility of the Siguniangshan granite ranges from 0.0077 - 0.0161 SI with an average value of 0.0123 SI. Most sedimentary formations are usually nonmagnetic. In general, the outcropping Proterozoic intermediate-felsic intrusive rocks and Triassic-Jurassic granite commonly have high magnetic susceptibility.




**Table 1.** Statistics of rock magnetic susceptibility in the LFZ

| Formation/Period | lithologic | min | max | average | Point No. |
|---|---|---|---|---|---|
| KD | quartz diorite | 0.0226 | 0.0562 | 0.0407 | D04 |
| KD | quartz diorite | 0.0016 | 0.0542 | 0.0238 | D09 |
| KD | quartz diorite | 0.0144 | 0.0868 | 0.0487 | D10 |
| KD | granite | 0.0001 | 0.0029 | 0.0005 | D05 |
| KD | granite | 0.0013 | 0.0082 | 0.0041 | D06 |
| KD | granite | 0.0009 | 0.0048 | 0.0032 | D07 |
| KD | granite | 0.0135 | 0.0359 | 0.0247 | D08 |
| KD | granite | 0.0015 | 0.0224 | 0.0082 | D11 |
| PG | granite | 0.0001 | 0.0008 | 0.0002 | D15 |
| PG | Serpentine (Not in-suit) | 0.0490 | 0.1203 | 0.0765 | D15-1 |
| Devonian | Mica schist | 0.0071 | 0.0840 | 0.0250 | D12 |
| Permian | Basalt (Not in-suit) | 0.0062 | 0.0710 | 0.0361 | D16 |
| Triassic | granite | 0.0460 | 0.0239 | 0.0118 | D02 |
| Triassic | granite | 0.0105 | 0.0259 | 0.0182 | D03 |
| Triassic | Syenite (Not in-suit) | 0.0122 | 0.0243 | 0.0201 | D13 |
| Jurassic | granite | 0.0077 | 0.0161 | 0.0123 | D14 |
| Cretaceous | Mudstone | 0.0003 | 0.0018 | 0.0011 | D01 |

### 3.2 Aeromagnetic and gravity data

The total field magnetic anomaly data used in this study are based on a compilation provided by the

China Aero Geophysical Survey and Remote Sensing Center for Natural Resources (Xiong et al., 2013, 2016), covering the LFZ and surrounding area. These magnetic data were derived from 47 different airborne surveys from 1958 to 2013 with different parameters, including line spacings and directions, altitude, and navigation methods. All data were adjusted to 1km above Earth's surface and sutured to be a continuous, merged dataset with a 1km grid (Fig. 2a). A detailed description of these datasets, surveys,

and data processing methodology can be found in the previous literature (Xiong et al., 2013). Such a merged dataset has been corrected by the International Geomagnetic reference field model to eliminate the influence of the various latitude by differential reduction to pole method (RTP) in the Oasis Motaj software (https:// www. seequent. com/ products-solutions/ geosoft–oasis-montaj). As a result, the processed anomalies show a slight northward shift in the RTP anomaly map (Fig. 2b). To remove the

effects of shallow magnetic bodies, the aeromagnetic data were continued upward to 20 km above the Earth's surface (Fig. 3a). In contrast, the first vertical derivative has been conducted to extract the anomalies derived from shallow magnetic bodies (Fig. 3b). The Bouguer gravity data used in this study



are based on a compilation provided by Development Research Center of China Geological Survey
(Zhang et al., 2011), covering the Longmenshan area with 1km grid (Fig. 4a). Such a dataset has been

removed the regional gravity field by first vertical derivative method.

### 3.3 2D forward modeling

The modeling of gravity and aeromagnetic data under the constraint of the velocity model produced
an integrated and less ambiguous result. He et al. (2017) inversed three S-wave velocity models using
the receiver function method based on the seismic data collected from 15 temporary seismological

stations and 42 permanent seismological stations. These models crossed the earthquake epicenters and
seismic gap show detailed velocity structure underneath the middle and southern segment of the LFZ.
To create a reliable structure model accommodating three types of geophysical data, three S-wave
velocity models were used to constrain the 2D magnetic and density modeling. The velocity model could
provide information of Moho depth. Then, the physical properties of different rocks were modeled, the

magnetic susceptibility contrast for the aeromagnetic data and the density for the gravity data. Meanwhile,
this method easily adds geological and structural information from previous studies and the
understanding of researchers during the modeling process.

The models were created using a 2D gravity and magnetic modeling package running on the Oasis
Montaj Program. Sequential gravity-magnetic modeling was performed by defining the density surface

in the crust according to the seismic models. The density values of the initial model referred to the
previous studies (Wang et al., 2014b; Zhang et al., 2014). The initial magnetic susceptibility used in the
modeling is a reference value from field observation to improve the interpretability of the model. For
example, the magnetic susceptibilities of basement rocks in the Sichuan Basin refer to the values of
outcropped Proterozoic intrusive rocks in the LFZ. Finally, the position, shape, dimensions, and contrast

of the physical properties of the rocks were adjusted to obtain the best fit between the observed and
calculated data.

### 3.4 3D inversion

To study the deep structure of the magnetic sources, we converted the aeromagnetic anomaly grid
data into a subsurface susceptibility model. In this study, we applied 3D inversion of the aeromagnetic





data (Li and Oldenburg, 1996), which has been widely used in geophysical exploration to create

quantitative models of ore-related magnetic sources and deep structure (Aitken and Betts, 2009; Fullagar

et al., 2004; Lü et al., 2013; Oldenburg et al., 1997; Roy and Clowes, 2000; Silva et al., 2001; Wang et

al., 2020a, b).

This inversion method assumes that the magnetic anomaly is caused by induced magnetization and

there is no remnant magnetization. It discretizes the subsurface space into many rectangular cells with

unknown susceptibilities. The inversion problem is formulated as an optimization problem. This method

minimizes a trade-off between data misfit and a model norm subject to a positivity constraint. The

objective function of this method is

$$Minimize \ \phi = \parallel W_d(d - Gm) \parallel_2^2 + \parallel W_m(m - m_0) \parallel_2^2$$

$$Subject \ to \ m \geq 0$$

where **μ** is a regularization parameter, **d** is the observed data, **m** is the model, **G** is the sensitivity

matrix, **W$_d$** is a diagonal data weighting matrix whose diagonal elements are reciprocals of estimated

noise standard deviations, and **W$_m$** is a model weighting matrix that consists of a weighted sum of zeroth-

and first-order finite difference matrixes. **m$_0$** is a reference model.

**4. Results**

**4.1 Major features of aeromagnetic data**

The Sichuan Basin is mainly characterized by NE-trending positive and negative banded magnetic

anomalies of a large scale and high intensity (Fig. 2a). The positive anomalies are mostly 200-400 nT,

and the negative anomaly is -300 nT. These features are different with the SGB and Yajiang Basin

displaying low magnetic anomalies with intensities of 10-60 nT. The LFZ is characterized by a gradient

zone of positive and negative anomalies on the ΔT image that is different from the feature of Xianshuihe

fault and sutures in this area (Fig. 2a). After reduction to the pole, the negative anomalies disappeared

because of the elimination of oblique magnetization, and the LFZ moved above the banded positive

anomalies (Fig. 2b). After 20 km of upward continuation, the regional magnetic field is separated into

two parts. The Sichuan Basin shows a large-scale high magnetic block in contrast to the broad and low





magnetic in the SGB (Fig. 3a).

As a transition zone between the Sichuan Basin and SGB, the LFZ is characterized by a boundary of two magnetic anomaly areas, but it is mainly located in the magnetic anomaly area of the Sichuan Basin. This result strongly supports that the LFZ thrusts above the basement of the Sichuan Basin, and the magnetic boundary between the Sichuan Basin and the SGB is probably located northwest of the Wenchuan-Maoxian fault ($F_1$). More importantly, the fault zone can be divided into three segments according to the magnetic anomaly feature along the strike with the boundary of the Siguniangshan-Dayi and Gucheng-Wulian (I and II in Fig. 2a). The middle and southern segments are characterized by positive magnetic anomalies and associated negative anomalies on the aeromagnetic ΔT image, but the magnetic anomaly features are discontinuity separated by the boundary of the Siguniangshan-Dayi. The northern segment shows a small-scale linear magnetic anomaly zone on the negative background field. The division of magnetic anomaly features is consistent with the segmentation based on the geological observation (Li et al., 2008). The seismic gap is located in the discontinuous area between the middle and southern segments. The Wenchuan earthquake and the Lushan earthquake occurred on the edge of two banded magnetic anomaly belts belonging to the middle and southern segments respectively.

In addition, there is an obvious magnetic discontinuous distributed from Daofu and Danba to Chengdu, which separates two distinct magnetic anomaly areas (Fig. 2a). It is inferred that there is a concealed fault named Daofu-Chengdu fault. The fault is divided into two segments by the Longmenshan fault zone. The western segment is distributed along Daofu - Qiaoqi and merges into the Xianshuihe fault in the northwest. The eastern segment started at Xiling, passed through Chengdu, and ended by the Longquanshan fault, which is characterized by an EW-trending linear discontinuity of magnetic anomalies extending approximately 90-100 km (Fig. 3b). it shows a small magnetic anomaly change after first vertical derivative because the anomalies of the shallow structure were highlighted in the magnetic field. The Longquanshan fault ($F_6$) does not change the magnetic anomalies in the central Sichuan Basin, which suggests that it is a shallow structure responding to the thrusting and napping of the LFZ.



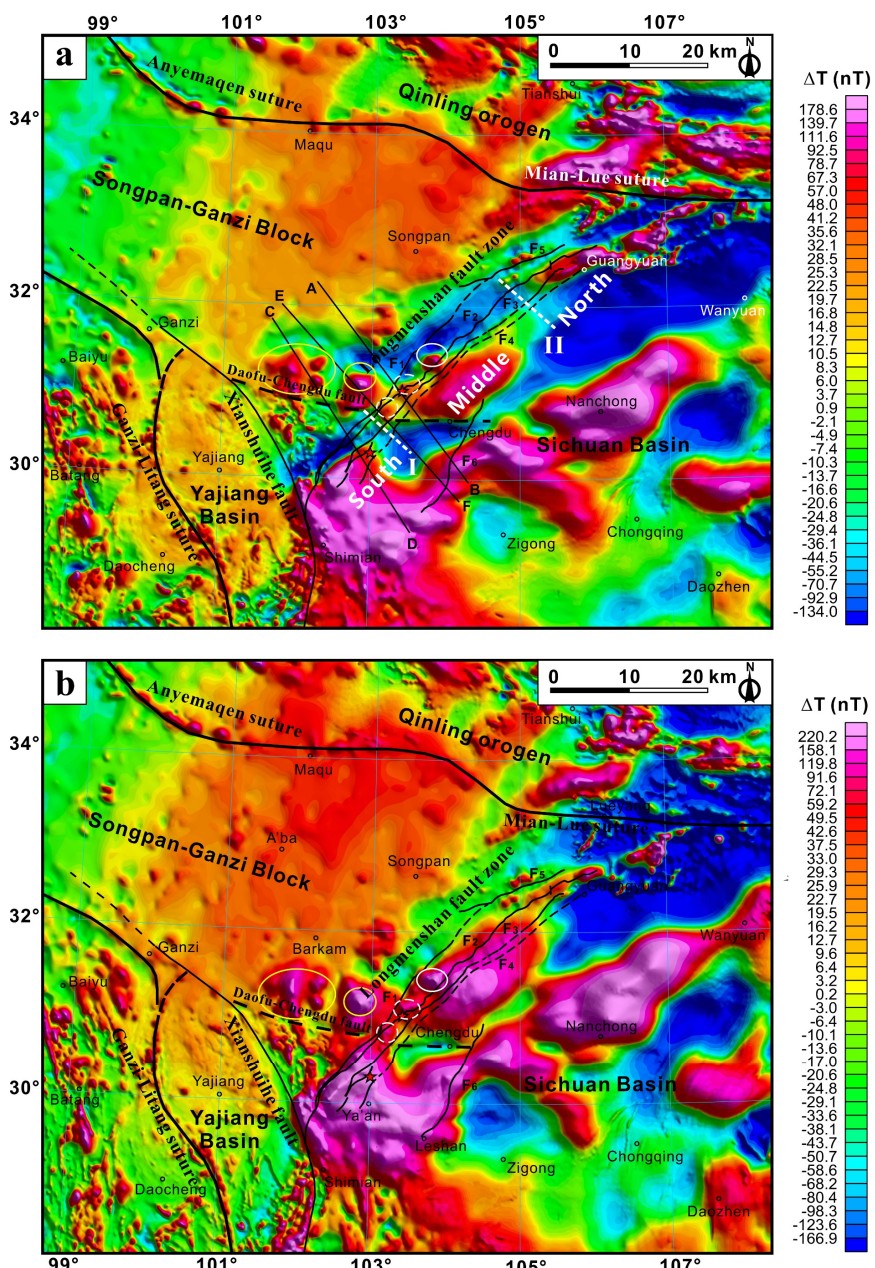

**Figure 2.** Aeromagnetic anomaly feature of the LFZ and adjacent area. (a) Aeromagnetic ΔT anomaly image. (b) Reduction to the pole (RTP) image of the aeromagnetic ΔT data. I: boundary of Siguniangshan-Dayi; II: boundary of Gucheng-Wulian.





**Figure 3.** Potential field transformations of RTP aeromagnetic ΔT anomaly data. (a) 20 km upward continuation of RTP aeromagnetic ΔT data. (b) First vertical derivative of RTP aeromagnetic ΔT data.



## 4.2 Major features of gravity data

According to the Bouguer gravity anomaly image (Fig. 4a), the Sichuan Basin has a high Bouguer

gravity anomaly with values ranging from -185 to -90 mgal. The SGB has a relatively low Bouguer

gravity anomaly with values ranging from -435 to -250 mgal. The feature illustrates the crust of the

Sichuan Basin has a higher density than that in the SGB. As a transition zone between two blocks, The

LFZ is characterized by a gradient zone with Bouguer gravity values of -290 to -185 mgal. From the first

vertical derivate of the Bouguer gravity anomaly map, a NE-trending high gravity anomaly belt extends

along the LFZ, which is probably caused by the high-density rocks in the upper crust. However, the

gravity anomaly belt is discontinuous along the LFZ, which could be divided into two segments with the

boundary of Siguniangshan-Dayi (I in Fig. 4a and 4b). The Bouguer gravity value of the middle segment

is -250~-185 mgal, while that of the southern segment is -290~-215 mgal. Moreover, the first vertical

derivate of Bouguer gravity anomaly in the middle segment is wider than that in the southern segment.

The density differences along the LFZ might be caused by structural heterogeneities in the crust. The

epicenter of the Wenchuan earthquake was located in a relatively high gravity anomaly area with Bouguer

gravity values ranging from -215 to -225 mgal, while the Lushan earthquake was located in an area with

low values of -250 to -260 mgal. The seismic gap is a transition zone between the middle and southern

segments with obviously lateral changing of gravity anomaly feature.

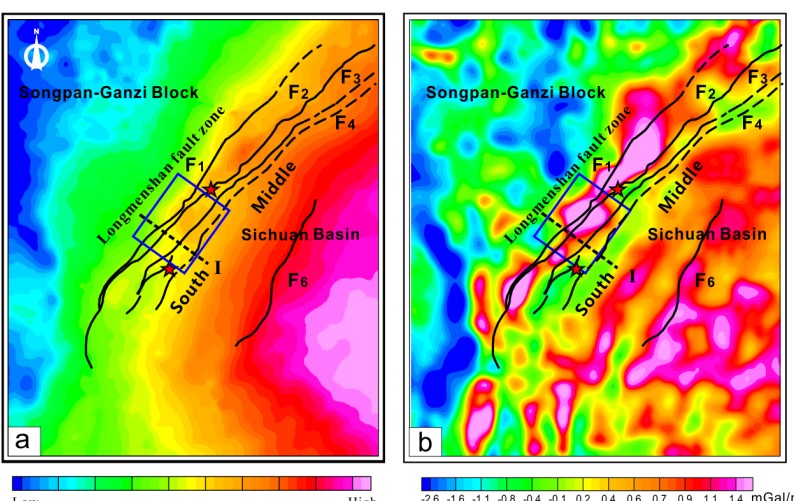

**Figure 4.** (a) Bouguer gravity anomaly image of the Longmenshan fault zone and surrounding areas. (b) first vertical
derivate of Bouguer gravity anomalies in the Longmenshan and adjacent areas. I: boundary of Siguniangshan-Dayi.



### 4.3 Magnetic and gravity modeling

**4.3.1 Profile AB**

Profile AB passes through the epicenter of the Wenchuan earthquake (Fig. 5). The gravity anomaly feature show obvious change on both sides of the LFZ, and the value is slightly increased in the Longmenshan area caused by the uplift of a high-density geological body. The model shows high magnetic susceptibility and density in the crust of the Sichuan Basin, ranging in 0.0189 - 0.0440 SI and

2.65 - 2.75 g/cm³, respectively. The depth to the top of the magnetic basement is 5 - 7 km, and the thickness is approximately 15 - 20 km. The thickness of the magnetic basement is stable in the Sichuan Basin, but it gradually thins and disappears beneath the northwest of LFZ. In contrast, the crust of SGB shows low magnetic susceptibility and density, ranging in 0.0013 - 0.0019 SI and 2.68 - 2.70 g/cm³, respectively. The depth to the top of the magnetic basement is approximately 4 - 10 km, and the thickness

is 12 - 18 km. There is a nonmagnetic area between the two basements with low density, which is a low-velocity zone extending downward to the crustal low-velocity zone in the Vs model (He et al., 2017). This zone is probably a weak and brittle area formed by the collision of the Sichuan Basin and the SGB.

The model indicates that the crust of the Sichuan Basin has two high magnetic blocks (HBM-1 and HBM-2 in Fig. 5). The HBM-1 lies in the central Sichuan Basin with an average magnetic susceptibility

of 0.0314 - 0.0440 SI, which corresponds to the NE-trending positive magnetic anomaly belt. The HBM-1 dips to the southeast. The HBM-2 has an average magnetic susceptibility of 0.0189 SI and contact with the overlying high magnetic block in the northwest. Compared with the location of the Wenchuan-Maoxian fault on the earth's surface, the magnetic basement of the Sichuan Basin obviously extends to the west of the Wenchuan-Maoxian fault with a distance of approximately 33 km. This result is consistent

with the resistivity model inversed from magnetotelluric data and the velocity model from deep seismic reflection profile (Guo et al., 2013; Zhu et al., 2008). The Wenchuan earthquake and its aftershocks were distributed inside the magnetic basement of the Sichuan Basin.



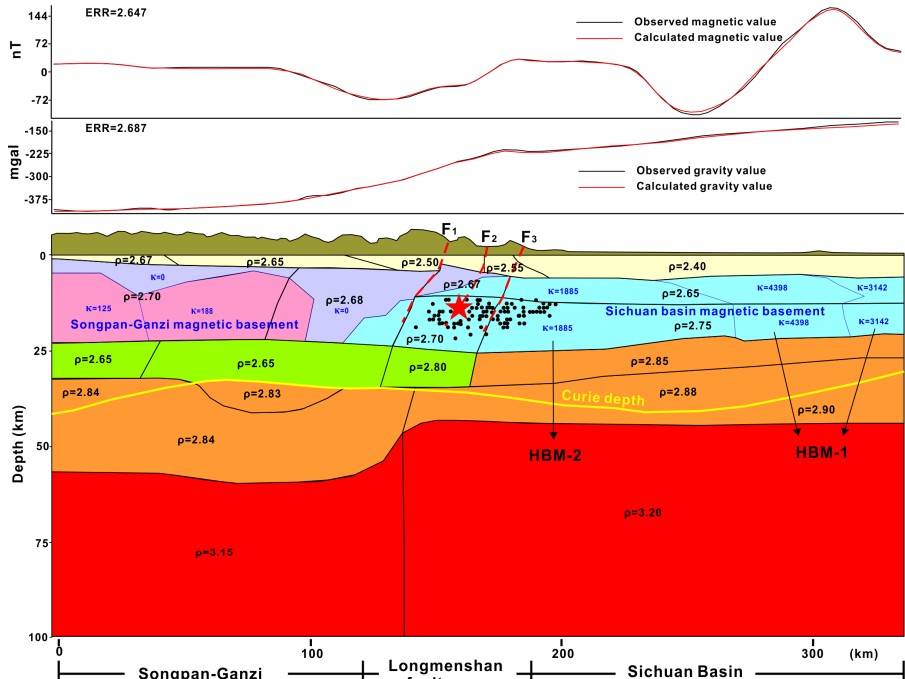

**Figure 5.** Two-dimensional gravity and magnetic model and interpreted crustal structure along profile AB. ρ marks the density in units of g/cm³, and κ marks the magnetic susceptibility in units of SI. The black line is the boundary of the density block, and the blue line is the boundary of the magnetic block.

### 4.3.2. Profile CD

Profile CD passes through the epicenter of the Lushan earthquake (Fig. 6). In contrast to profile AB, the Bouguer gravity anomaly doesn't show obvious changes on both sides of the LFZ. The values only slightly increase in the Longmenshan area, which is caused by the upward thrusting of high-density geological bodies. However, the magnetic anomaly values show large changes on both sides of the LFZ. The values are high in the Sichuan Basin and decrease rapidly in the LFZ. Therefore, the model shows that the basement of the Sichuan Basin has high magnetic susceptibilities of 0.0250 - 0.0440 SI. The density is 2.65 - 2.70 g/cm³. The magnetic layer dips to the northwest. The depth to the top of the magnetic basement is approximately 5 - 11 km, and the thickness is approximately 17 - 23 km. In the west of the Wenchuan – Maoxian fault ($F_1$), there is a magnetic body with a moderate magnetic susceptibility of 0.0126 SI. The depth to the top of the magnetic body is 3 - 6 km, and the thickness is 16 - 20 km. The magnetic body is inferred to be intermediate-felsic intrusive rock because there are many



Triassic and Jurassic granite, syenite, and granodiorite plutons outcropping on the surface with high

magnetic susceptibility. The basement of the SGB has a low magnetic susceptibility of 0.0013-0.0038

SI. The depth to the top of the magnetic basement is approximately 4 - 7 km, and the thickness is

approximately 13 - 16 km.

     In this model, the magnetic basement of the Sichuan Basin extends approximately 19 km west of

the Wenchuan – Miaoxian fault. And, the magnetic basement subducts beneath the LFZ at a high angle

rather than a low angle in profile AB. The top of the magnetic basement shows large fluctuations, which

suggests that the crystalline basement is highly thrusted and deformed in the southern segment of the

LFZ. Moreover, there is no double-layer magnetic structure in this profile. The Lushan earthquake and

its aftershocks are distributed in the rigid magnetic basement of the Sichuan Basin.

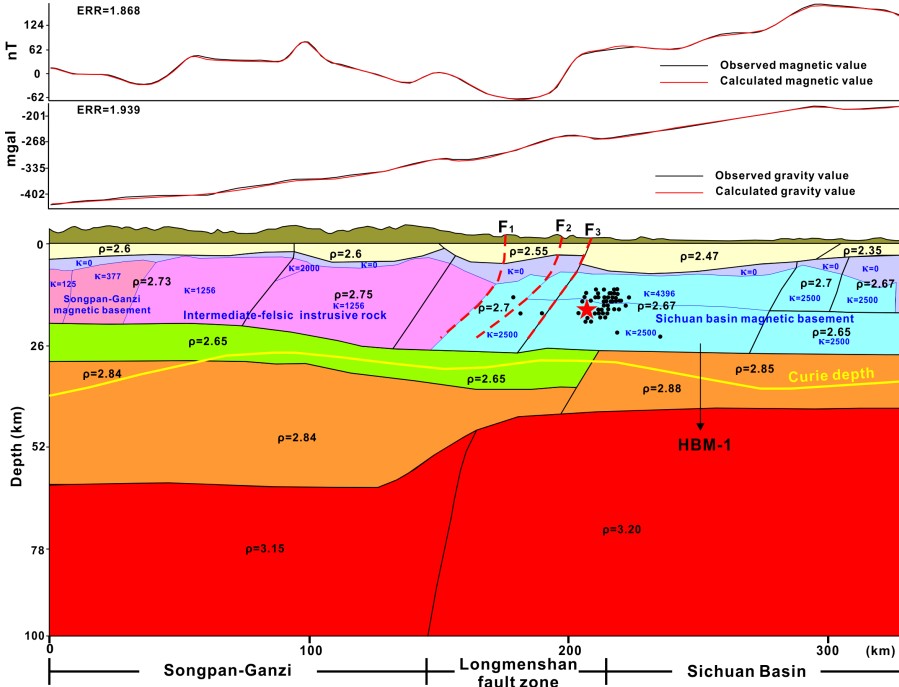


**Figure 6.** Two-dimensional gravity and magnetic model and interpreted crustal structure along profile CD. ρ marks
the density in units of g/cm³, and κ marks the magnetic susceptibility in units of SI. The black line is the boundary
of the density block, and the blue line is the boundary of the magnetic block.

### 4.3.3. Profile EF

Profile EF passes through the seismic gap (Fig. 7). The Bouguer gravity anomaly shows the same




pattern as profile AB, which displays obvious changes with the boundary of the LFZ. The basement of

the Sichuan Basin has two high magnetic blocks as in profile AB. The HBM-1 is distributed in the central

Sichuan Basin with a magnetic susceptibility of 0.0151 - 0.0377 SI. The depth to the top of the magnetic

basement is 4 - 10 km, and the thickness is approximately 15 - 19 km. The HBM-2 is locally distributed

beneath the Longmenshan area with a magnetic susceptibility of 0.0126 SI. In this model, the HBM-1

doesn't directly in contact with the HBM-2 and dips to the northwest. A large area of nonmagnetic

sedimentary cover or basement is distributed between the two high magnetic blocks. In the northwest of

LFZ, A high magnetic anomaly peak is modeled by a pluton with a magnetic susceptibility of 0.0126 SI,

corresponding to the Siguniangshan granite. The pluton extends downward to the middle-upper crust

from the surface, and the thickness is approximately 22 km. Particularly, unlike the previous two profiles,

the magnetic basement under the Longmenshan area does not have a complex thrust and nappe structure.

And, the magnetic basement subducts approximately 17 km west of the Wenchuan-Miaoxian fault. The

low-density layer in the middle and upper crust does not extend below the magnetic basement in the

Sichuan Basin.

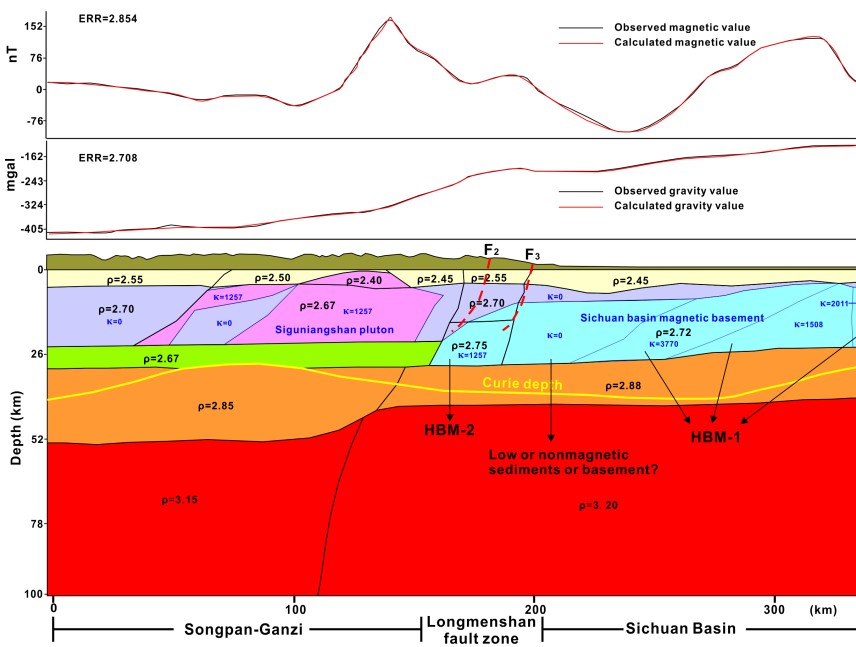


**Figure 7.** Two-dimensional gravity and magnetic model and interpreted crustal structure along profile EF. ρ marks
the density in units of g/cm³, and κ marks the magnetic susceptibility in units of SI. The black line is the boundary
of the density block, and the blue line is the boundary of the magnetic block.



### 4.4 3D inversion of aeromagnetic data

330 To make the deep structure of the LFZ visible, the subsurface space was divided into 280 × 256 × 200 cells forming a cube volume of 840 km× 768 km× 100 km. After 3D inversion, a susceptibility value for each cell was obtained. Cells with high susceptibility provide a good representation of magnetic sources for the magnetic highs. To observe the magnetic sources visually, Fig. 8 displays the inverted susceptibility model by an isosurface with a high susceptibility value of 0.015 SI. We set the bottom

335 depth of the susceptibility model as 50km because the curie depth of this area is 45-50km (Xiong et al., 2016). The inversion model shows that the isosurface delineates two high magnetic blocks under the Sichuan basin (HMB1 and HMB2 in Fig. 8). The Wenchuan and Lushan earthquake lies on the different magnetic block. The seismic gap is characterized by a low magnetic area between two high magnetic blocks. It is worth noting that the magnetic block is uplifted to shallow surface beneath the LFZ which

340 is well-matched with the outcropping of the Baoxing and Pengguan complex on the ground. The susceptibility model proved the strong crust shortening and basement deformation of the Sichuan basin.

 The results of the inversion are shown by depth slices in the range of 10–40 km (Fig. 9). Figures 9a and 9b display that the high magnetic rocks are distributed in a small area beneath the LFZ. When the depth increases to 30 and 40km, two NE-trending high magnetic areas show a large scale in the Sichuan

345 basin. Two earthquakes lie in high magnetic areas from 20 to 40 km. The seismic gap is absent of high magnetic rocks. The vertical slices of the inversion model along profiles AB, CD, and EF are shown in Fig. 10. The Wenchuan and Lushan earthquakes are near or in the high magnetic blocks, but the seismic gap shows a weak magnetic area which is the southeast end of the HMB1(Fig. 10a, b, c).



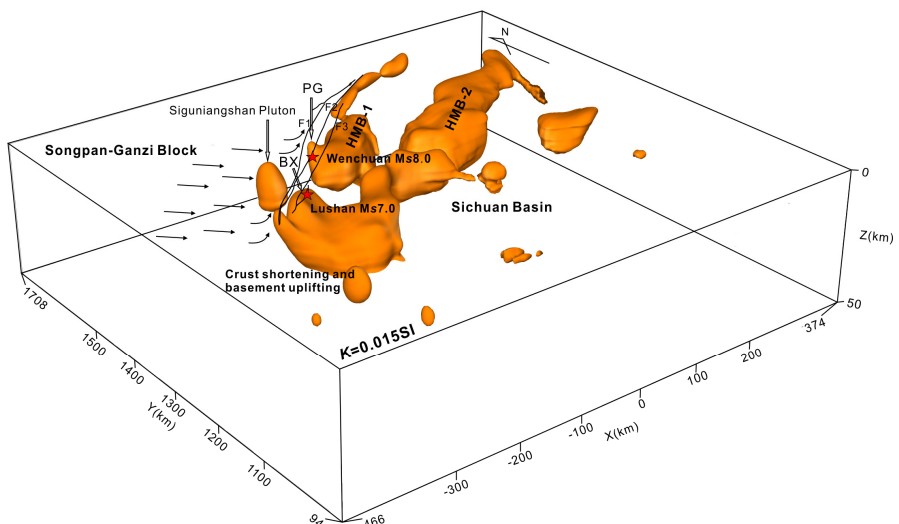

350    **Figure 8. Three-dimensional susceptibility model of Longmenshan fault zone. HMB-1 and HMB-2: two high magnetic blocks in Sichuan basin.**

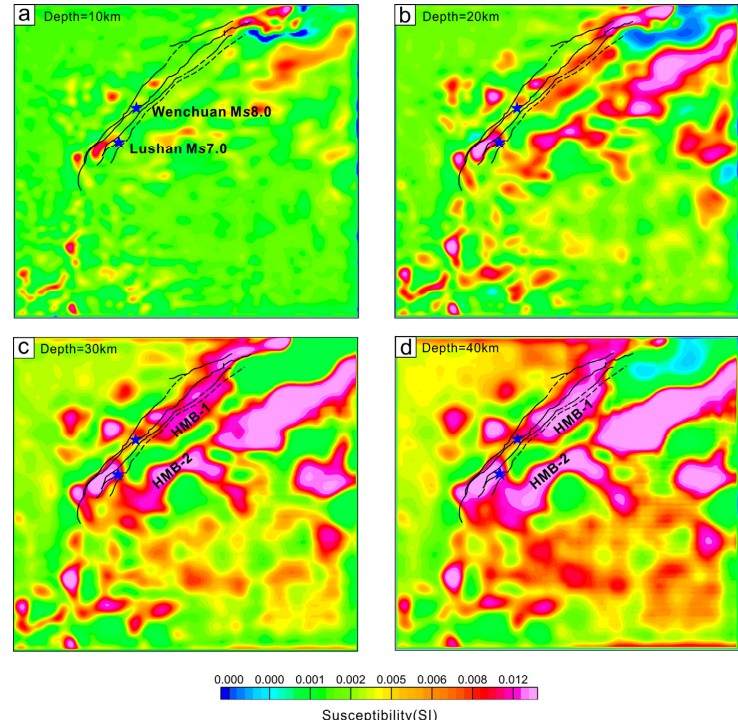

**Figure 9.** Horizontal slices of the inverted susceptibility model at different depths from 10–40 km every 10 km.



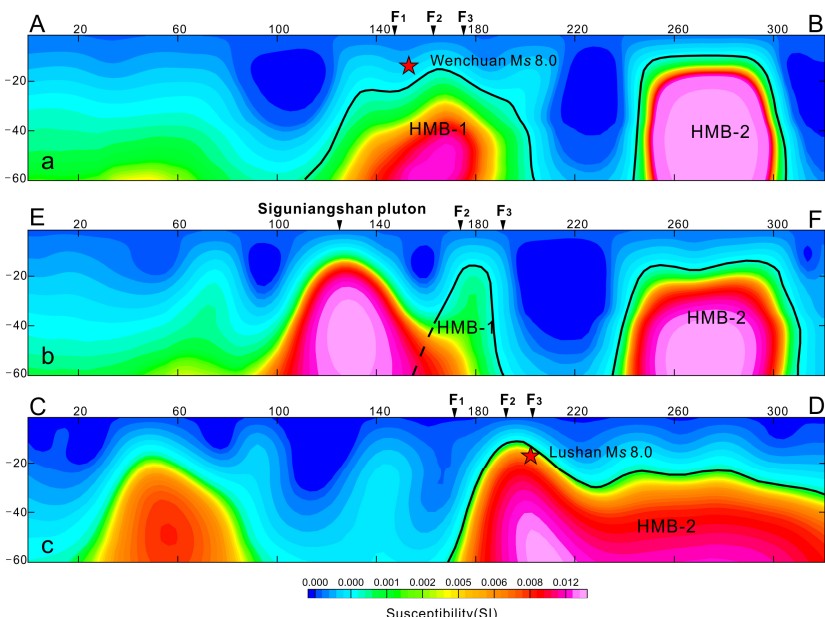

**Figure 10.** Vertical slice of the inversion model along line AB, CD and EF. Its location is marked in Fig. 2a.

## 5. Discussion

### 5.1 The source of aeromagnetic anomaly

To understand the deep structure of the LFZ, geophysical data should be explained comprehensively including aeromagnetic and gravity anomalies. Regionally, the LFZ shows similar magnetic and gravity anomaly feature to the Sichuan basin. The inversion models indicate that the basement of the Sichuan Basin has extended to the west of the Wenchuan-Maoxian fault and reached the deep of SGB. The results agreed with the previous conclusion that the LFZ has been thrust above the Sichuan Basin during the Late Indosinian-Early Yanshanian (Guo et al., 2013; Wang et al., 2015; Xiong et al., 2016; Zhang et al., 2010; Zhu et al., 2008). The convergent process caused partial melting of the crust and formed a series of intermediate-acidic intrusive rocks in the SGFB. The isotopic and chronology data show that these rocks have Proterozoic clastic zircon cores and Nd model ages ($T_{DM}$), which indicate that there is Proterozoic Yangtze-type continental crust beneath the Songpan-Ganzi area (Dai et al., 2011; Hu et al., 2005; Zhao et al., 2007a, b).

To find the source of the aeromagnetic anomaly, we are trying to analyze outcropped rock



assemblage in the western margin of the Sichuan Basin. The outcropped Precambrian complexes, such as Kangding, Baoxing, and Pengguan, are well-matched with the high magnetic anomalies (Fig. 11). According to field observations of magnetic susceptibility, the quartz diorite, granite from Kangding Complex, and the Serpentine from Pengguan Complex usually have high magnetic susceptibility. These

Precambrian rock assemblages could produce large-scale and high magnetic anomalies. The Triassic-Jurassic Syenite and granite have moderate magnetic susceptibility in the SGB producing local magnetic anomalies. Based on the understanding of previous geological surveys, the basement is mainly composed of Neoarchean-Paleoproterozoic high-grade metamorphic rocks that are able to produce high magnetic anomaly in the Sichuan Basin, such as Kangding Group (SBGMR, 1991; Xiong et al., 2016).

Recently, a large amount of geochronological and geochemical evidence has shown that the Kangding complex has arc signatures, representing metamorphic products of Neoproterozoic arc-related acidic plutons rather than Neoarchean and Paleoproterozoic crystalline basement (Chen et al., 2005; Du et al., 2007; Geng et al., 2007; Kang et al., 2017; Lai et al., 2015; Liu et al., 2009; Zhou et al., 2002). Meanwhile, zircon U–Pb data have shown that the Huangshuihe and Yanjing Groups also formed in the

Neoproterozoic, and the E'bian Group formed in the late Mesoproterozoic (Chen et al., 2018; Du et al., 2005; Ren et al., 2013). Moreover, granite and rhyodacite collected from the drill core of hydrocarbon exploration in the central Sichuan Basin have isotopic age of 701.5-794 Ma (Gu et al., 2014; Luo et al., 1986). These widespread Meso-Neoproterozoic rift-related magmatism and sedimentation records imply that Sichuan Basin probably played a key role in the assembly and breakup of the Rodinia supercontinent

(Cui et al., 2015). Therefore, the banded positive magnetic anomaly is probably related to Meso-Neoproterozoic magmatic events rather than the presence of rigid Neoarchean and Paleoproterozoic crystalline basements in the Sichuan Basin. The 2D and 3D models show two high magnetic blocks indicating there are probably two concealed Meso-Neoproterozoic igneous rock belts distributed in the Sichuan Basin (Fig. 12). These large-scale igneous rock belts formed the rigid blocks of the Sichuan

Basin which subducted beneath the SGB and apt to accumulate stress.



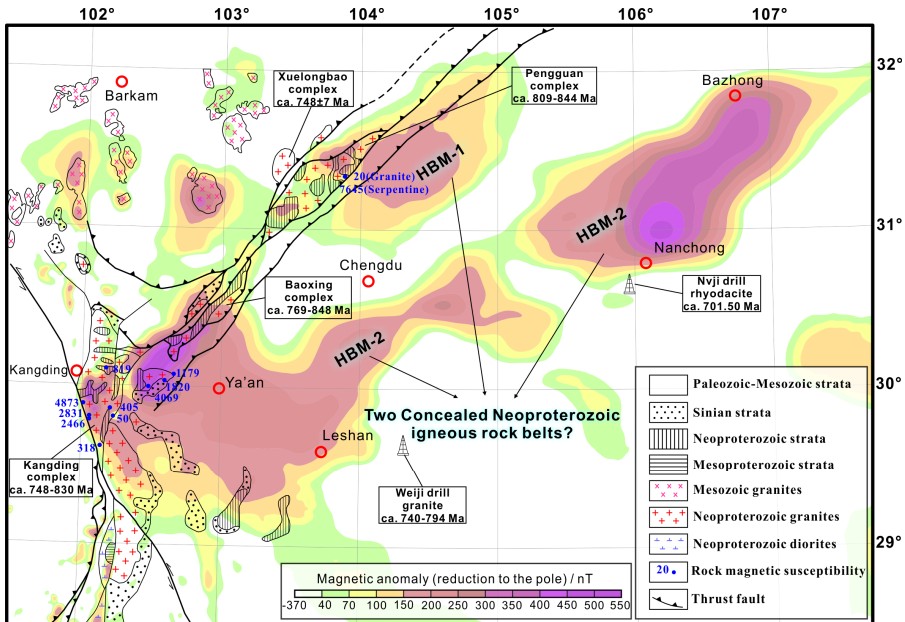

**Figure 11.** RTP aeromagnetic ΔT anomaly contours and outcrops of Precambrian basement and intrusive rocks on the western margin of the Sichuan Basin (modified after Du et al. (2014) and Geng et al. (2007)).

## 5.2 The crustal structure of the LFZ

The gravity and magnetic anomalies show obvious lateral changes along the strike of the LFZ, which can be divided into southern, middle, and northern segments (Fig. 2a and 4). The southern segment is characterized by the magnetic anomaly gradient zone and the low Bouguer gravity anomaly, while the middle segment has a magnetic anomaly gradient zone and high Bouguer gravity anomaly. The Wenchuan and Lushan earthquakes were distributed in the middle and southern segments respectively. The northern segment is characterized by a negative magnetic field with some of the linear magnetic anomaly zone superimposed. The division of aeromagnetic and gravity anomaly features is the same as the surface deformation segmentation of the LFZ (Li et al., 2008). Therefore, the deep structure reflected by aeromagnetic and gravity anomaly probably controlled the evolution and deformation of the LFZ.

The 2D and 3D inversion of multi-sourced geophysical data suggests that the basement of the Sichuan Basin is heterogeneous in magnetic and density (Fig. 8, 9, 10, and 12). There are two NE-trending high magnetic blocks with different scales and intensities. The Wenchuan earthquake lies on the HBM-1 with a small scale and low intensity. The Lushan earthquake lies on the HBM-2 with a large



scale and high intensity. The seismic gap is characterized by a low or non-magnetic area between two
blocks. According to the 2D models, the magnetic basement subducts beneath the LFZ with a distance
of approximately 33 km west of the Wenchuan-Maoxian fault in profile AB. However, the distances are
17 km and 19 km in profiles CD and EF, respectively. Therefore, the subducted distance of the basement
has a large lateral change along the western margin of the Sichuan Basin. The basement beneath the
middle segment of the LFZ wedges farther than that under the southern segment, forming a "stair-shaped"
along the LFZ (Fig. 12).

The 2D models indicate that the two disastrous earthquakes and their aftershocks were mainly
distributed in the magnetic basement of the Sichuan Basin (Fig. 9). Moreover, the 3D susceptibility model
shows an obvious uplift beneath the LFZ suggesting the magnetic basement has undergone a strong
deformation (Fig. 8 and 10). The deformation of the basement is also proved by the highly shortened
crystalline crust of the Yangtze block shown by the deep seismic reflection profile (Guo et al., 2013).
Meanwhile, the thickness gradually decreases when the magnetic basement wedges into the eastern
Tibetan Plateau. These characteristics indicate that the crust of the Sichuan Basin has undergone strong
elastic shortening. The seismic images show that the crustal low-velocity zone extends beneath the
epicenters of the Wenchuan and Lushan earthquakes (He et al., 2017). Therefore, the occurrence of the
two earthquakes may be closely related to the destruction of the magnetic rigid basement through the
detached upper crust of the SGB collided with the crust of the Sichuan Basin.



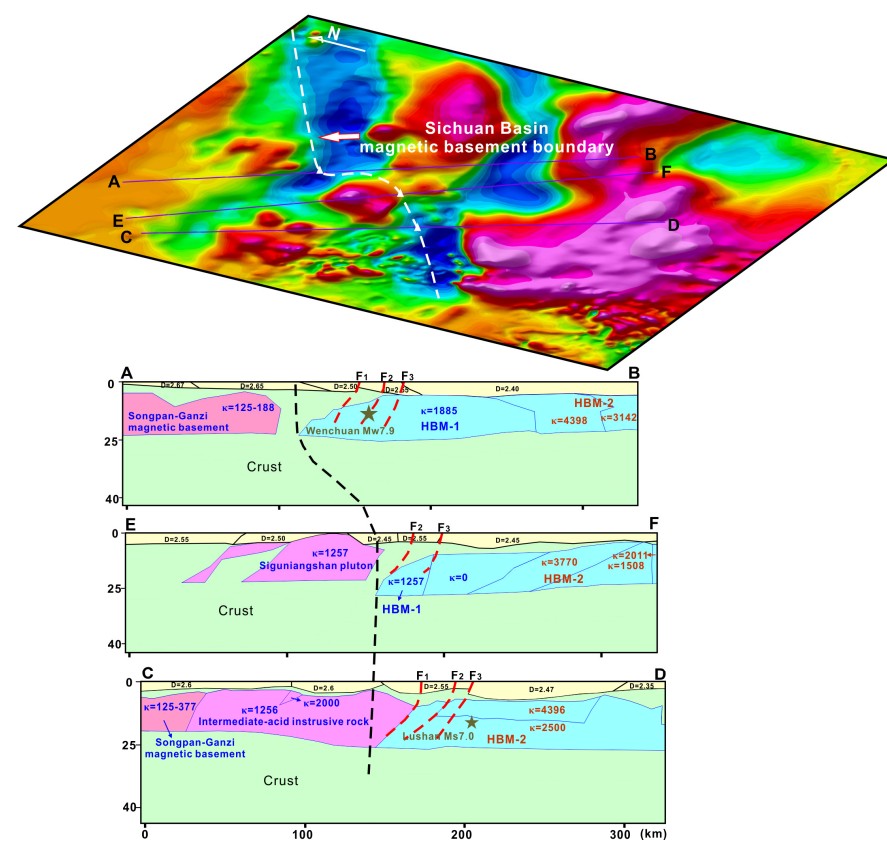

**Figure 12. Western margin of the magnetic basement in the Sichuan Basin.**

**5.2 Seismogenesis mechanism of the LFZ**

The Wenchuan earthquake occurred in the Yingxiu-Beichuan fault, which is part of the central-
front range fault system in the middle segment of the LFZ. The Lushan earthquake occurred in a blind
reverse fault east of the Shuangshi-Dachuan fault, which is part of the front range fault system in the
southern segment of the LFZ. Although the two segments are separated by seismic gaps with small
distances, the geological deformation is quite different. The front-range structure in the southern segment

is much more complicated than that in the middle segment. Meanwhile, the range of the latest structural
deformation increases from 30 km in the middle segment to 150 km in the southern segment (Xu et al.,
2013). The focal mechanism shows that the Lushan earthquake was a pure thrust event without obvious
ruptures on both sides (Chen et al., 2013). The axis of the maximum horizontal stress is oriented NW-SE





(Luo et al., 2015). However, the Wenchuan earthquake was dominated by thrusts with a dextral strike-slip component. The surface rupture mainly extends northeast with a distance of approximately 300 km. The axis of the maximum horizontal stress occurs in several different directions (Luo et al., 2015). These features indicate that the deformation mechanisms of the middle and southern segments are different. High-resolution geodetic data show that the surface deformation has obvious changes on both sides of the seismic gap (Wang et al., 2011). Therefore, this study proposes that the differences in the focal mechanism of the two earthquakes may be closely related to the differential thrusting mechanism caused by basement heterogeneity in the western margin of the Sichuan Basin.

Two earthquakes and their aftershocks occurred in the rigid magnetic basement of the Sichuan Basin, which is characterized by high-velocity areas in seismic images (Wang et al., 2015). There is a seismic gap with low Vp and Vs, high Poisson's ratio, and high conductivity between the two earthquakes. It is inferred to be a fluid-rich ductile crust extending to the middle and lower crust (Pei et al., 2014; Wang et al., 2015; Zhan et al., 2013). Consequently, compared with the magnetic basement under the earthquake epicenter, there is no high magnetic basement beneath the seismic gap. The inferred Daofu-Chengdu fault crosses the seismic gap. The fault could be divided into two segments. The western segment may be related to a tear zone and accompanied by the intrusion of intermediate-acid rocks. The S-wave velocity models indicate that there is an NW-trending Moho uplift from the seismic gap to the northwest, which is a fault zone that caused mantle upwelling and partial melting (He et al., 2017; Liang et al., 2018). The eastern segment of the Daofu-Chengdu fault cuts the basement of the Sichuan Basin and extends to Longquanshan Mountain. The sedimentary covers are commonly nonmagnetic in the Sichuan Basin and cannot cause changes in magnetic anomalies. The magnetic anomaly decays obviously after 20 km upward continuation, which suggests that the displacement is small on both sides of the fault. The fault may be closely related to the early thrust in the southern segment of the LFZ and the uplift of Longquanshan Mountain.

This study proposes a schematic model (Fig. 13). The rigid basement of the Sichuan Basin subducts beneath the Songpan-Ganzi fold belt during the Late Indochina-early Yanshanian. The basement beneath the middle and northern segments of the LFZ extends further than under the southern segment. The lateral change in the basement caused the Songpan-Ganzi fold belt to tear into two pieces with different



geodynamic systems (Slab 1 and Slab 2 in Fig. 13). Different geodynamic systems lead to different activities of the two slabs that form a tear zone for the emplacement of intermediate-acid intrusive rocks. This kind of mechanism is also found in the southern Tibetan Plateau which is characterized by the

tearing and dischronal subduction of the Indian continental slab (Hou et al., 2006, 2011). With the continuous uplift of the Tibetan Plateau, the compression stress is increasing in the western margin of the Sichuan Basin. In 2008 and 2013, two earthquakes with different focal mechanisms occurred successively in the middle and southern segments of the LFZ. These two earthquakes were produced by two slabs thrust above the "stair-shaped" magnetic basement. Therefore, the Wenchuan earthquake

occurred in the middle and northern segments of the LFZ, and the Lushan earthquake was restricted to the southern segment. Apparently, the central Sichuan Basin may be involved in the early thrust process of the southern segment of the LFZ, which is represented by the displacement of the basement and formation of the Longquanshan fault ($F_6$).

The geodynamic process of the two earthquakes is summarized as follows:

(1) With the uplift of the Tibetan Plateau, the Songpan-Ganzi block continued to move toward the southeast and collided with the basement of the Sichuan Basin. Compression stresses were prior accumulation in the middle-northern segment because the basement beneath the middle and northern segment wedges farther than under the southern segment. Finally, the Wenchuan Ms 7.9 earthquake was triggered in the middle segment of the LFZ. Due to different geodynamics between the middle and

southern segments of the LFZ and the constraints of the tear zone and the Siguniangshan pluton, no large-scale surface rupture occurred to the southwest of the Wenchuan earthquake. However, the rupture extended approximately 340 km toward the northeast.

(2) The compression stress was completely released in the middle and northern segments of the LFZ after the Wenchuan earthquake occurred. The stress shifted to the southern segment and finally

triggered the Lushan Ms7.0 earthquake. Due to the constraints of the Yangtze block's irregular crust and the Xianshuihe fault on both sides, the Lushan earthquake was a pure thrust event without obvious rupture on both sides.



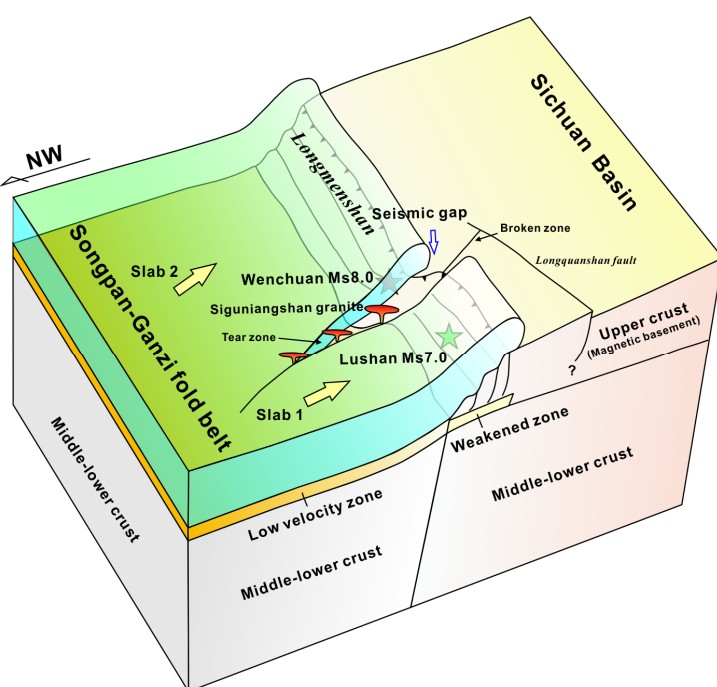

**Figure 13.** Schematic deep structure model of the Longmenshan fault zone and adjacent areas.

**6. Conclusions**

The aeromagnetic and gravity anomaly revealed significant variations in the physical property of the crust along the LFZ. The anomaly feature of the LFZ could be divided into three segments that are consistent with the surface deformation. 2D forward modeling and 3D inversion obtained a comprehensive magnetic and density model in the LFZ and adjacent area. The result suggests that two high magnetic blocks of the Sichuan Basin are subducted beneath the LFZ, which is inferred as Meso-Neoproterozoic igneous rock belt apt to accumulate stress. The Wenchuan earthquake lies on a small scale and moderate magnetic susceptibility block while the Lushan earthquake lies on a large scale and high magnetic susceptibility block. There is a low or non-magnetic area that is void aftershocks for both earthquakes. The lateral change of magnetic and gravity anomaly indicates structural heterogeneity along the LFZ. Moreover, the magnetic basement beneath the middle segment wedges further and has a lower dip angle than under the southern segment. The magnetic basement involves more intense deformation beneath the epicenter of the two earthquakes than under the seismic gap. Due to the irregular shape of

the basement and the constraints of the tear zone and intermediate-acid intrusive rocks, the thrust

mechanism is different in the middle and southern segments of the LFZ. It provides an essential tectonic

framework for the genesis of the two earthquakes with different focal mechanisms. This study provides

new geophysical evidence for mapping the deep structure of the LFZ.

**Data availability**

Data will be made available on request. The use of the Oasis Montaj software is authorized by

Beijing Maiqin Nengyuan Jishufuwu Co., Ltd. (https: //www. seequent. com / products – solutions /

geosoft – oasis - montaj/).

**Author contribution**

Hai Yang was responsible for the idea and methodology of the study, field investigation, compilation

and analysis of data, and writing the paper. Shengqing Xiong was responsible for the acquisition and

management of financial support, contribution to experimental design. Qiankun Liu and Fang Li

investigated and revised the ideas of the article. Zhiye Jia and Xue Yang processed the aeromagnetic and

gravity data. Haofei Yan and Zhaoliang Li was responsible for the 3D inversion. All authors contributed

to the review of the manuscript.

**Competing interest**

The authors declare that they have no known competing financial interests or personal relationships

that could have appeared to influence the work reported in this paper.

**Acknowledgements**

We thank Professor Chuntao Liang and Fujun He from Chengdu University of Technology for

providing seismic imaging results and enthusiastic help. Thanks to all of our colleagues for their hard

work. This research was supported by the National Natural Science Foundation of China grant

(U2244220) and China Geological Survey Project grant (DD20190551, DD20230351). Thank you to the

anonymous reviewers for their hard work and constructive comments.



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
