# Peer review of "The crustal structure of the Lomgmenshan fault zone and its implications for seismogenesis: New insight from aeromagnetic and gravity data"

_EGUsphere, 2023_

## Referee Comment (RC2)

Review report of manuscript entitle '**The crustal structure of the Lomgmenshan fault zone** and its implications for seismogenesis: New insight from aeromagnetic and gravity data (egusphere-2023-1119)' submitted by Hai Yang, Shengqing Xiong, Qiankun Liu, Fang Li, Zhiye Jia, Xue Yang, Haofei Yan, and Zhaoliang Li

The present manuscript aims to investigate the deep structure of the seismic gap in the Longmenshan fault zone (LFZ) based on 2D forward modeling and 3D inversion of aeromagnetic and Bouguer gravity data. The authors have made significant efforts to establish a correlation between the deep crustal structure and seismic activity. However, I would like to offer some suggestions that could further enhance the manuscript.

- Several previous studies have already investigated the deep structure of the LFZ using seismic data, and authors also briefly mentioned these details in the introduction. It would be valuable to elaborate on the new insights derived from this manuscript and provide a detailed comparison with the findings of these previous studies. This comparative analysis can help readers better understand the novel contributions of the current research
- I failed to understand how authors carried out the magnetic modelling, especially the zig-zag blocks considered with different magnetic susceptibility values. Does it have geological significance, or were they considered only to achieve the best fit? Moreover, I was wondering why the magnetic models weren't extended to the Curie depth.
- Previous studies reveal a low velocities (Vp, & Vs) in the seismic gap region, whereas high low velocities (Vp, & Vs) were imaged below the Lushan and Wenchuan hypocenter. Surprisingly, the crustal density models presented in the manuscript do not exhibits such heterogeneities. I'm curious about the nature of the faults (F1-F3) depicted in the models and why there doesn't appear to be any density variation across them.
- The authors suggest that the aeromagnetic and gravity anomalies reveal significant variations along the LFZ, which is consistent with surface deformation. I agree that magnetic anomaly characteristics differ between the seismic gap and seismicity regions. However, I could not find contrasting gravity signatures across these regions. What I see from Fig. 4 is only an NE-SW trending gravity high running nearly parallel to the LFZ, with some minor variations in amplitudes.
- A detailed description of the aeromagnetic datasets, such as line spacings and sampling intervals, should be included in the manuscript. Otherwise small-scale structures interpreted from magnetic anomaly maps, especially in the seismic gap region have no meaning. I would also suggest that the authors show the location of flight lines on one of the magnetic anomaly maps.
- The paper employs 2D forward modeling based on aeromagnetic and Bouguer gravity data for studying deep crustal structure. However, it would be beneficial to provide more details regarding how the authors have incorporated seismic velocity data to constrain the crustal models. Specifically, for profiles AB and EF, where distinct

bipolar magnetic anomalies are observed, it would be helpful to have a proper justification for how the authors fitted these anomalies.

• Similarly, there is limited information provided about the 3D gravity inversion. It would be helpful to discuss how the regularization parameter ( $\mu$ ) and weighting matrix were selected. Additionally, it's unclear whether in-situ measurements were used to constrain the susceptibility values of the model. To enhance the clarity of the manuscript, consider including contour plots depicting the misfit between the observed and calculated anomalies, as well as the root mean square (RMS) error for each iteration.

**Minor comments:**

- Although the overall language and presentation of the paper are clear, but some sentences could benefit from rephrasing for improved readability. I quoted a few here:
- Could you please clarify what you mean by 'visible magnetic and density model beneath the LFZ?
- .....structure heterogeneities are widely distributed be-neath the LFZ.... Do you mean to say structure is heterogenous across the LFZ.
- The earthquake epicenters show high magnetic anomalies and the edge of high Bouguer gravity anomalies ...... accumulate stress.. This sentence is difficult to follow and may rephrased. It might be clearer to say that earthquake epicenters are often located in regions with high magnetic anomalies and gravity gradient .....
- The LFZ was attacked by two different earthquakes....Need to be rephrased
- In Figure 1b, it could be beneficial to include seismicity data. Different colors could be used to represent earthquakes at various focal depths. Consider implementing this suggestion in other figures as well.
- Adding seismicity data to the aeromagnetic and gravity anomaly maps (Figures 2 to 4) would provide readers with a more detailed understanding of the correlation between anomalies and seismic activity

---

## Author Comment (AC2)

**The crustal structure of the Lomgmenshan fault zone and its implications for seismogenesis: New insight from aeromagnetic and gravity data**

MS No.: egusphere-2023-1119
MS type: Research article

**Reply to Reviewer Comments (RC2)**

Comment from Authors to Reviewer 2

We thank Reviewer 2 for their very constructive comments. We believe that the comments and changes based on them have completed and improved the manuscript and we will certainly use the suggestions for future work. The detailed replies are listed in the supplement file.

Kind regards,
Hai Yang
On behalf of the authors

**General comments:**

- Several previous studies have already investigated the deep structure of the LFZ using seismic data, and authors also briefly mentioned these details in the introduction. It would be valuable to elaborate on the new insights derived from this manuscript and provide a detailed comparison with the findings of these previous studies. This comparative analysis can help readers better understand the novel contributions of the current research.

  The previous studies proposed several models based on seismic and MT data to analyze the deep physical structure around the seismic gap, such as the ductile deformation area, fault zone, or fluid-bearing ductile flow. Pei et al. (2014) suggest that the aftershock gap is weak and the ductile deformation is more likely to occur in the upper crust within the gap under the near NW-SE compression. Liang et al. (2018) propose that the fault-parallel stress difference across the gap may imply that the gap is the point where the crust of the eastern Tibetan plateau is being torn apart, resulting in the upwelling of hot mantle materials to produce partial melting in the lower crust. Wang et al. (2014c) argued that fluid extrusion from the lower crust and upper mantle led to the existence of ductile bodies that played important roles in earthquake generation. These models commonly indicate the presence of ductile zones or partial melting under the gap area. Meanwhile, the segmentation of deep structure and surface deformation along the fault zone has been published in many literature. However, there are many problems that still need further discussion, such as the genesis of the ductile zone or partial melting. what kind of dynamic drives this process? And why the Wenchuan and Lushan earthquakes are so different?

In this manuscript, we create an integrated 2D and 3D magnetic and density model to discuss the deep structure of the Longmenshan fault zone. The advance of this model is considering the basement shape of the Sichuan Basin which plays an important role in the geodynamic process of the Longmenshan fault zone. The Sichuan Basin has two NE-trending banded high magnetic blocks extending beneath the LFZ that firmly support the crust of the Sichuan Basin was downward subduction toward the LFZ. More importantly, the basement subducts to approximately 33 km west of the Wenchuan-Maoxian fault with a low dip angle beneath the middle segment of the LFZ, whereas the distance decreases to approximately 17 and 19 km under the southern segment. Therefore, the subducted distance of the basement has a large lateral change along the western margin of the Sichuan Basin. The basement beneath the middle segment of the LFZ wedges farther than that under the southern segment, forming a "stair-shaped" along the LFZ. The irregular basement shape possibly causes stress differences between the middle and southern segments that formed a tear zone with low velocity and high conductivity. Generally, the structural heterogeneity of the fault zone leads to different geodynamic features and seismogenic mechanisms.

- I failed to understand how authors carried out the magnetic modelling, especially the zig-zag blocks considered with different magnetic susceptibility values. Does it have geological significance, or were they considered only to achieve the best fit? Moreover, I was wondering why the magnetic models weren't extended to the Curie depth.

Yes. It has geological significance. Actually, there are two main factors that affect the shape of the magnetic anomaly curve during the fitting process, magnetic susceptibility and the shape of fitting geological bodies. If one of the factors is constrained by geological information, a model with geological significance can be obtained. Otherwise, the modeling results will be multiplicity for geological interpretation. In this manuscript, the initial magnetic susceptibility was given by the measured data. Two kinds of rocks have high magnetic susceptibility producing the high magnetic anomaly along the fitting profiles. Proterozoic quartz diorite has a moderate magnetic susceptibility of 0.0238 - 0.0487 SI with an average value of 0.0377 SI, while Proterozoic granite has values of 0.0002 - 0.0247 SI with an average value of 0.0068 SI. The Triassic and Jurassic granites are widely distributed in the west of the LFZ, and their magnetic susceptibility ranges from 0.0118 - 0.0201 SI, and their average value is 0.0167 SI. The magnetic susceptibility of the Siguniangshan granite ranges from 0.0077 - 0.0161 SI with an average value of 0.0123 SI. In the fitting process, the magnetic susceptibility of basement rock in the Sichuan basin referred to the outcropped Proterozoic intrusive rocks in the LFZ. The HMB1 refers to the magnetic susceptibility of Proterozoic quartz diorite ranging from 0.0238 – 0.0487 SI. The HMB2 refers to the magnetic susceptibility of Proterozoic granite ranging from 0.0002 – 0.0247 SI. The Siguniangshan granite uses an average value of 0.0123 SI.

The curie depth is an ideal surface in the Earth's crust where the ferromagnetic minerals lose their magnetic property. It is a reference for the bottom of magnetic bodies. During the fitting process, we gave a specific magnetic susceptibility to each geological body and made the model easy to interpret. However, the value may be

biased from the geological facts. Therefore, the bottom of the magnetic body cannot match the Curie surface.

It is worth noting that no matter how the fitting parameters change, the boundary of the Sichuan Basin basement will not change, because gravity and magnetic data are sensitive to tectonic boundaries.

- Previous studies reveal a low velocities (Vp, & Vs) in the seismic gap region, whereas high low velocities (Vp, & Vs) were imaged below the Lushan and Wenchuan hypocenter. Surprisingly, the crustal density models presented in the manuscript do not exhibits such heterogeneities. I'm curious about the nature of the faults (F1-F3) depicted in the models and why there doesn't appear to be any density variation across them.

First, the Bouguer gravity anomaly map shows comprehensive density variation from Moho to the Earth's surface. Due to the effect of the Moho depth, the density variation of the shallow crust is not obvious in this image. The first vertical derivate of Bouguer gravity anomaly enhances the information from the shallow crust. The low-density felsic intrusive rocks show low gravity anomaly in this image, such as the Siguniangshan pluton. Both of them show obvious changes between the southern and middle segments. Second, the gravity anomaly feature shows obvious change on both sides of the LFZ in the middle segment (profile AB and EF in Fig.4 and Fig. 6), and the value is slightly increased in the Longmenshan area caused by the uplift of a high-density geological body. However, the gravity anomaly feature doesn't show obvious change on both sides of the LFZ in the southern segment (profile CD in Fig.5). If we cut through a profile along the Longmenshan fault zone, the gravity anomaly shows a remarkable change (Fig. S1). In our model, the average density value is slightly decreased from the middle segment to the southern segment. However, the models provide rough density information for deep structure, because the accurate modeling result needs more constraints of geological data. Third, the scale of gravity data is 1:200000 and compiles with 1km × 1km grid, which is not enough to analyze the detail of the faults. Especially, the density of rocks does not vary as much as magnetic susceptibility. Therefore, gravity anomaly shows the feature of the fault system, rather than the density variation of faults (F1-F3).

[Figure]

Figure S1. The Bouguer gravity anomaly profile cutting through the Wenchuan and Lushan earthquakes.

- The authors suggest that the aeromagnetic and gravity anomalies reveal significant variations along the LFZ, which is consistent with surface deformation. I agree that magnetic anomaly characteristics differ between the seismic gap and seismicity regions. However, I could not find contrasting gravity signatures across these regions. What I see from Fig. 4 is only an NE-SW trending gravity high running nearly parallel to the LFZ, with some minor variations in amplitudes.

The Bouguer gravity anomaly map shows comprehensive density variation from Moho to the Earth's surface. The first vertical derivate of Bouguer gravity anomaly enhances the information from the shallow crust. Both of them show obvious changes between the southern and middle segments. The Bouguer gravity value of the middle segment is -250 ~ -185 mgal, while that of the southern segment is -290 ~ -215 mgal. The first vertical derivate of the Bouguer gravity anomaly in the middle segment is wider than that in the southern segment. The density differences along the LFZ might be caused by structural heterogeneities in the crust. For example, the Triassic limestone usually has high density, so the thrust of the Triassic limestone could produce obvious gravity anomaly along the LFZ. The linear gravity anomalies in the Sichuan basin are the presence of the thrust fault belt, such as the Longquanshan fault belt. The gravity anomaly is not the same as the aeromagnetic anomaly, because the density of rocks does not vary as much as magnetic susceptibility.

- A detailed description of the aeromagnetic datasets, such as line spacings and sampling intervals, should be included in the manuscript. Otherwise small-scale structures interpreted from magnetic anomaly maps, especially in the seismic gap region have no meaning. I would also suggest that the authors show the location of flight lines on one of the magnetic anomaly maps.

Agreed. The description of the aeromagnetic datasets is added in the manuscript. There are three kinds of data used in the study (Fig. S2). The line spacings of 1:500, 000-1:1000,000 aeromagnetic datasets are 5 and 10 km respectively. The sampling interval is 10-15m. The line spacings of 1:100, 000-1:200,000 aeromagnetic datasets are 1 and 2 km respectively. The sampling interval is 10-15m. The line spacing of 1:50,000 aeromagnetic datasets is 500m. The sampling interval is 5-10m.

[Figure]

Figure S2. The scale of aeromagnetic dataset in the study area.

- The paper employs 2D forward modeling based on aeromagnetic and Bouguer gravity data for studying deep crustal structure. However, it would be beneficial to provide more details regarding how the authors have incorporated seismic velocity data to constrain the crustal models. Specifically, for profiles AB and EF, where distinct bipolar magnetic anomalies are observed, it would be helpful to have a proper justification for how the authors fitted these anomalies.

Sequential gravity-magnetic modeling was done first by defining the depths to the upper, middle, and lower crust and Moho discontinuities from the seismic image. The density values of the initial model referred from the previous density structure in the Longmenshan area (table S1) and were constrained by the seismic velocity results (Wang et al., 2014b; Zhang et al., 2014). The initial magnetic susceptibility was given by the measured data. The magnetic susceptibility of basement rock in the Sichuan basin referred to the outcropped Proterozoic intrusive rocks in the LFZ. Then the position, shape, dimensions, and physical property contrast of the basement rock were adjusted to get the best fit between the observed and calculated data.

Table S1 Crustal density structure of Longmenshan area

| Location | Sichuan Basin | Longmenshan | Songpan-Ganzi |
|---|---|---|---|
| Sedimentary cover | 2.34 | 2.67 | 2.46 |
| Upper crust | 2.61 | 2.67 | 2.59 |
| Lower velocity layer | | | 2.55 |

| Middle crust | 2.75 | 2.79 | 2.71 |
|---|---|---|---|
| Lower crust | 2.89 | 2.82 | 2.82 |
| Mantle | 3.3 | | 3.25 |

- Similarly, there is limited information provided about the 3D gravity inversion. It would be helpful to discuss how the regularization parameter (μ) and weighting matrix were selected. Additionally, it's unclear whether in-situ measurements were used to constrain the susceptibility values of the model. To enhance the clarity of the manuscript, consider including contour plots depicting the misfit between the observed and calculated anomalies, as well as the root mean square (RMS) error for each iteration.

  The regularization parameter (μ) is following the traditional method. In 3D magnetic inversion, there are two fitting processes including data fitting and model fitting. The regularization parameter is the ratio of the maximum value of data fitting to the maximum value of model fitting. In addition, The Generalized Cross Validation (GCV) method is also used to calculate the regularization parameters. The inversion results are basically consistent with the traditional method. The weighting matrix uses Depth weighting. The measured susceptibility isn't used to constrain the inversion model. The default setting for each iteration is stopping the iteration if the error is less than 0.001.

**Specific comments:**

- Could you please clarify what you mean by 'visible magnetic and density model beneath the LFZ?

  Yes. The expression changed to "Based on the compiled aeromagnetic data and Bouguer gravity data, we have tried to create a more detailed and reasonable magnetic and density model using 2D forward modeling and 3D inversion and made the deep structure of the LFZ visible."

- ⋯structure heterogeneities are widely distributed be-neath the LFZ…. Do you mean to say structure is heterogenous across the LFZ.

  Yes. The expression changes to "The research shows that structure is heterogenous across the LFZ."

- The earthquake epicenters show high magnetic anomalies and the edge of high Bouguer gravity anomalies ……. accumulate stress. This sentence is difficult to follow and may rephrased. It might be clearer to say that earthquake epicenters are often located in regions with high magnetic anomalies and gravity gradient …..

Agreed. The sentence changes to "The earthquake epicenters are located in regions with high magnetic anomalies and gravity gradients that are associated with rigid blocks where apt to accumulate stress."

- The LFZ was attacked by two different earthquakes….Need to be rephrased

Yes. The sentence changes to "Two different earthquakes happened in the LFZ within a short time and the risk of the seismic gap has challenged Earth scientists."

- In Figure 1b, it could be beneficial to include seismicity data. Different colors could be used to represent earthquakes at various focal depths. Consider implementing this suggestion in other figures as well.

Thank you. We are trying to add the seismicity data in Figure 1b (see below), but it is hard for the reader to get geological information. Therefore, we don't make this change in the manuscript.

[Figure]

Figure 1. Geological background of the LFZ.

- Adding seismicity data to the aeromagnetic and gravity anomaly maps (Figures 2 to 4) would provide readers with a more detailed understanding of the correlation between anomalies and seismic activity

Agreed. We add the seismicity data in Figures 2 and 4.

[Figure]

Figure 2. Aeromagnetic anomaly feature of the LFZ and adjacent area. (a) Aeromagnetic ΔT anomaly image. (b) Reduction to the pole (RTP) image of the aeromagnetic ΔT data. I: boundary of Siguniangshan-Dayi; II: boundary of Gucheng-Wulian. The blue dots are earthquakes with focal depth less than or equal to 5km. The pink dots are earthquakes

with focal depth 5-10km. The black dots are earthquakes with focal depth 10-15km. The orange dots are earthquakes with focal depth 15-20km. The green dots are earthquakes with focal depth greater than 20km.

[Figure]

Figure 4. (a) Bouguer gravity anomaly image of the Longmenshan fault zone and surrounding areas. (b) first vertical derivate of Bouguer gravity anomalies in the Longmenshan and adjacent areas. I: boundary of Siguniangshan-Dayi. The blue square is seismic gap. The blue dots are earthquakes with focal depth less than or equal to 5km. The pink dots are earthquakes with focal depth 5-10km. The black dots are earthquakes with focal depth 10-15km. The orange dots are earthquakes with focal depth 15-20km. The green dots are earthquakes with focal depth greater than 20km.

---

## Author Response (AR1)

**Dear Editor, dear Referees,**

Thank you for providing very helpful comments and suggestions on our manuscript. We are pleased to report that we have followed the vast majority of your recommendations and feel that this has significantly improved the readability and presentation of the paper.

Two referees are concerned about the difference between the results of current research and the findings of previous research. Actually, we are trying to create the deep structure of the LFZ from the perspective of magnetic and density, because the aeromagnetic and gravity data covers the LFZ and adjacent two blocks. It is good for the deep structure study of the convergent zone and is sensitive to lateral change of the tectonic boundary. In this study, integrated 2D and 3D models suggest that the downward subducted basement of the Sichuan Basin is irregular in shape and heterogeneous in magnetic and density. The different focal mechanisms of the two earthquakes and the genesis of the seismic gap may be closely related to the differential thrusting mechanism caused by basement heterogeneity in the western margin of the Sichuan Basin. We hope this research could provide new evidence for the study of the deep structure and geodynamic process of the LFZ.

The following document provides responses to each point raised in both reviews, including a description of the changes made to the manuscript and figures. A document with highlighted changes to the manuscript is also attached.

With best regards, on behalf of all the authors,

Hai Yang

**Referee Comment #1**

Based on the compiled aeromagnetic data and Bouguer gravity data, the study used 2D forward modeling and 3D inversion method to create a more detailed and visible magnetic and density model beneath the Longmenshan Fault Zone. There are plenty of seismic studies in this region, but aeromagnetic and Bouguer gravity studies are rare, therefore this study has its value to further understand the tectonic background of the Wenchuan and Lushan earthquakes. I recommend to be published with minor revisions listed below.

**Specific comments:**

• Line 83: "Triassic syntectonic adakitic-type granitoids are widely distributed in the SGB, which are likely sourced from the partial melting of an underlying Proterozoic basement that is part of the Sichuan Basin". The Sichuan basement may extend further west beyond the LFZ, but I don't think it can account for the "syntectonic adakitic-type granitoids" far west from the LFZ.

Agreed. The syntectonic adakitic-type granitoids with Mid-proterozoic TDM age (1.23-1.44Ga) and some geochemical indicators suggest that the Songpan-Ganzi terrane is underlain with Yangtze-type basement (Zhao et al., 2007). Recently, the SinoProbe-02 deep seismic reflection profile suggests the crust of the Yangtze crystalline crust probably extends beneath the easternmost Tibetan Plateau to the Longriqu fault (west of the Longriba fault). However, the evidence is still insufficient to prove the syntectonic adakitic-type granitoids from the partial melting of the Sichuan basement. Therefore, we change the sentence to "Triassic syn-tectonic adakitic-type granitoids are widely distributed in the SGB, which are likely sourced from the partial melting of an underlying Proterozoic Yangtze-type crystalline basement". See Line 83.

• Figure 1: add descriptions for the black lines (profiles?), pink squares (measurement points?) to the captions.

Yes. The descriptions add to the captions. The blue line is the seismic profile from He et al. (2017). The pink squares are magnetic susceptibility measurement points. See Line 95.

• Figure 2b, 3a, 3b actually show that the anomaly across F6 (Longquan fault) are different. On Figure 3b, the values are positive (>3) and negative (